# An Integrated Gather-and-Distribute Mechanism and Attention-Enhanced Deformable Convolution Model for Pig Behavior Recognition

**DOI:** 10.3390/ani14091316

**Published:** 2024-04-27

**Authors:** Rui Mao, Dongzhen Shen, Ruiqi Wang, Yiming Cui, Yufan Hu, Mei Li, Meili Wang

**Affiliations:** 1College of Information Engineering, Northwest A&F University, Yangling 712100, China; maorui@nwafu.edu.cn (R.M.); dongzhencs@nwafu.edu.cn (D.S.); 2021013067@nwafu.edu.cn (R.W.); 2916227225@nwafu.edu.cn (Y.C.); huyufan@nwafu.edu.cn (Y.H.); limei@nwsuaf.edu.cn (M.L.); 2Shaanxi Engineering Research Center of Agriculture Information Intelligent Perception and Analysis, Yangling 712100, China

**Keywords:** pig, behavior recognition, gather-and-distribute mechanism, multi-path coordinate attention, DM-GD-YOLO

## Abstract

**Simple Summary:**

The abnormal behavior of pigs can undermine their growth performance and economic value. Therefore, the precision and timeliness of behavioral recognition are crucial for maintaining pig health and advancing intelligent farming. This study introduces an innovative DM-GD-YOLO model, an optimized variant of YOLOv8, which integrates a deformable convolution module with enhanced multi-path coordinate attention and a gather-and-distribution mechanism. Through experiments conducted on a farm with about 30 pigs per pen, the results reveal that the proposed model can effectively recognize four common behaviors (walking, lying, sniffing, and kneeling) and three abnormal behaviors (fighting, mounting, and fence climbing) in pigs. Compared to traditional methods, the model exhibits superior performance and provides a practical solution for enhancing the welfare of pigs.

**Abstract:**

The behavior of pigs is intricately tied to their health status, highlighting the critical importance of accurately recognizing pig behavior, particularly abnormal behavior, for effective health monitoring and management. This study addresses the challenge of accommodating frequent non-rigid deformations in pig behavior using deformable convolutional networks (DCN) to extract more comprehensive features by incorporating offsets during training. To overcome the inherent limitations of traditional DCN offset weight calculations, the study introduces the multi-path coordinate attention (MPCA) mechanism to enhance the optimization of the DCN offset weight calculation within the designed DCN-MPCA module, further integrated into the cross-scale cross-feature (C2f) module of the backbone network. This optimized C2f-DM module significantly enhances feature extraction capabilities. Additionally, a gather-and-distribute (GD) mechanism is employed in the neck to improve non-adjacent layer feature fusion in the YOLOv8 network. Consequently, the novel DM-GD-YOLO model proposed in this study is evaluated on a self-built dataset comprising 11,999 images obtained from an online monitoring platform focusing on pigs aged between 70 and 150 days. The results show that DM-GD-YOLO can simultaneously recognize four common behaviors and three abnormal behaviors, achieving a precision of 88.2%, recall of 92.2%, and mean average precision (mAP) of 95.3% with 6.0MB Parameters and 10.0G FLOPs. Overall, the model outperforms popular models such as Faster R-CNN, EfficientDet, YOLOv7, and YOLOv8 in monitoring pens with about 30 pigs, providing technical support for the intelligent management and welfare-focused breeding of pigs while advancing the transformation and modernization of the pig industry.

## 1. Introduction

Over the past five years, pork constituted approximately 38% of global meat consumption, signifying its vital role in the human diet. García-Gudiño et al. [1] gathered data on consumer perceptions of pig production and animal welfare in Iberia, finding that consumers strongly prefer naturally raised free-roaming animals and value animal welfare greatly. Despite numerous policies, tools, and regulations aiming to enhance and safeguard the welfare of farmed animals, persistent welfare issues continue [2]. This underscores the need for innovative methods within animal production systems.

The understanding of pig behavior, particularly abnormal behaviors such as fence climbing and mounting, is a significant factor in their health and welfare assessment [3,4]. Conventionally, livestock management involves extensive labor and resources to manually monitor and record animal health and behavior patterns [5]. In contrast, real-time intelligent recognition of pig behavior considerably boosts efficiency and accuracy compared to manual methods [6,7]. This technique assists farm workers in quickly determining the physiological status of pigs, enabling immediate interventions and treatments when warranted. Such advances are crucial for optimizing pig farming efficiency and promoting animal welfare [8,9].

The ongoing progress in computer vision, pattern recognition, and deep learning technologies facilitates real-time surveillance and analysis of pig behavior using data sources like images and videos. This progress has led to the development of non-contact intelligent recognition methods for better animal welfare farming. Chen et al. [10] developed an algorithm integrating Xception and LSTM, achieving a 95.9% accuracy in recognizing pig feeding behavior. Liu et al. [11] utilized a combination of convolutional and recurrent neural networks in a computer vision-based approach for recognizing and locating tail-biting interactions among group-housed pigs. Riekert et al. [12] adopted a Faster R-CNN object detection pipeline and a neural architecture search-based network for feature extraction from pigs, achieving an 80.2% mean average precision (mAP) in recognizing pig position and posture. However, these models are not designed for widespread recognition of both individual and group abnormal behaviors in pigs. Additionally, they use traditional convolutional operations with fixed kernels for feature extraction, thus limiting their effectiveness due to non-rigid deformations of the targets [13,14]. Dai et al. [15] introduced deformable convolution networks (DCN) as a promising resolution to these challenges. However, DCN’s invariant distribution of sampling weights cannot adjust according to specific data attributes and task requirements, thus restricting its efficacy in recognizing complex pig interaction behaviors in real farming environments.

Currently, the YOLO-series models, one-stage detection models, have proven effective for pig behavior recognition due to their simple structure and appreciable accuracy [16,17]. Li et al. [18] improved the YOLOX model by incorporating a normalization attention mechanism, increasing the mAP to 92.23% in recognizing pig attack behaviors—5.3% higher than the baseline YOLOX model. Odo et al. [19] used the YOLOv4 and YOLOv7 to recognize pig ear-biting behaviors independently, achieving respective AP scores of 91.8% and 91.9%. In addition, Luo et al. [20] enhanced the YOLOv5 model by integrating the ECA-Net attention mechanism to amplify feature channel expressions, thereby achieving a 92.04% mAP in recognizing five pig behaviors like standing and kneeling. However, these models, despite being somewhat effective in pig behavior recognition, uniformly use Feature Pyramid Networks (FPNs) [21] or their equivalents to amalgamate multi-level features. This usage risks the inefficacy of multi-level feature utilization for pig behavior recognition tasks and potential information loss.

Therefore, the research problem lies in the limitations of the existing model in simultaneous recognition of multiple pig behaviors, particularly abnormal behaviors, and its deficiency in robust feature extraction and fusion capabilities, creating a predisposition to information loss. Addressing these challenges, the purpose of this study was to develop a proposal for a novel method to augment the feature extraction and fusion capabilities of the model. This will enable accurate recognition of both common and abnormal pig behaviors, aiding farmers in uncovering potential hidden risks in pigs and mitigating economic losses. By enhancing the DCN with the MPCA mechanism and integrating the C2f-DM module with the GD mechanism [22], the YOLOv8’s feature extraction and fusion capacities are bolstered. The proposed DM-GD-YOLO can accurately classify three abnormal behaviors (fighting, fence-climbing and mounting) and four common behaviors (sniffing, walking, lying, and kneeling), achieving an mAP of 95.3%. This model offers technical support for real-time monitoring of abnormal pig behaviors in farming environments, with the primary goal of optimizing pig farming efficiency and advancing animal welfare.

## 2. Materials and Methods

### 2.1. Dataset

#### 2.1.1. Data Sources

In the process of intensive pig breeding, farm workers not only need to obtain health information through the common behaviors of pigs, but also need to recognize and respond to abnormal behaviors of pigs in a timely manner. Therefore, we studied seven typical behaviors of pigs, including sniffing, lying, walking, kneeling, fighting, fence climbing, and mounting, among which fighting, fence climbing, and mounting are abnormal behaviors. The recognition rules of each behavior [23] are shown in Table 1.

Given the absence of comprehensive public datasets containing all seven types of pig behavior, we have developed a more encompassing dataset that includes multiple behaviors. This dataset is intended to fulfill the needs for research on the automatic recognition of complex pig behaviors. We used the HIKVISION DS-2CD3T25-I5 surveillance cameras (Hikvision, Hangzhou, China) installed on the online supervision platform of Linyou Zhengneng Agriculture and Animal Husbandry Technology Co., Ltd. in Shaanxi Province to collect videos of pigs’ behavior. Mounted 3.6 m above the ground, the cameras capture footage at a top angle of 30° within the pig houses numbered 2-3-1, 2-4-1, and 2-7-1 across the fattening facility, each housing about 30 pigs and having feed troughs. The cameras can export the video data directly, and the recorded videos can be obtained without interruption. The recorded pigs comprised three breeds, Landrace, Yorkshire, and crossbred pigs, totaling 153 pigs. The fattening pigs weighted between 30 kg and 110 kg, with an age range of 70 to 150 days.

The dimensions of the pen on the farm were 7.3 m × 5.0 m, providing an average usable area of 1.22 m^2^ per pig, exceeding the minimum movement area requirement of 1.0 m^2^ for a 110 kg pig. Installation of ventilation devices, such as exhaust fans, effectively mitigates the adverse effects of high humidity, condensation, and unexpected wind on the pigs. Pen temperature is maintained between 18 and 22 degrees Celsius to meet the environmental requirements for fattening pigs. Each pen is equipped with two nipple drinking fountains for free water access, meeting the standards of GB5749-2022 [24]. Automatic supply of dry feed is facilitated through two feed troughs, each featuring four feeding spaces. A head-and-shoulder barrier between each feeding space enables simultaneous accommodation of multiple pigs. The structure of the pen is shown in Figure 1.

To improve the robustness of the dataset, we captured videos of pigs’ behavior under different lighting conditions (noon with the most intense light, cloudy days with uniform light but low overall brightness, and indoor artificial lighting with relatively constant light intensity). Thanks to the advantages of non-contact data collection, none of the pigs used to collect the data were harmed or affected.

#### 2.1.2. Data Preprocessing

Data preprocessing has four processes including image acquisition, screening and filtering, labeling, and data enhancement.

Image acquisition. Python scripts were used to intercept images in jpg format every 12 frames of the collected videos.Screening and filtering. After excluding images with significant pig occlusion, 11,999 images were finally extracted. We then used filters to blur the noise spots in the image and reduce interference with model training.Labeling. We utilized the labelImg software (https://github.com/HumanSignal/labelImg (accessed on 17 March 2024)) to manually label the pig behaviors in the images. Throughout the process, pigs with occluded areas exceeding 30% or with occluded heads were not labeled. Adhering to this criterion, we produced standard txt format annotation files following the COCO dataset format.After labeling is completed, we divided the 11,999 images and annotation files into a training set, a validation set, and a test set at a ratio of 7:2:1. Since an image may contain multiple instances of behaviors, the number of instances for each behavior in the dataset is shown in Table 2.Data enhancement. Data enhancement techniques can increase the diversity of samples and improve the robustness of the model. In this study, the Mosaic Data Augmentation (MDA) [25] method in the YOLO model was deployed to augment the data through five methods, including image stitching, mirroring, clipping, random rotation, and HSV tone enhancement, such that the scale of the dataset used for model training was expanded fivefold.

### 2.2. Our Model

#### 2.2.1. DCN-MPCA

Pig behavior is characterized by its transience and a broad spectrum of postural shifts, often resulting in non-rigid deformation. This imposes significant challenges on behavior recognition tasks. Most existing models employ a conventional fixed convolution operation in their convolution modules, maintaining relatively static positions on the feature map. This approach inevitably leads to substantial information loss, compromised adaptability to diverse behaviors, and limited generalization ability. To address these shortcomings, we adopted Deformable Convolution Networks (DCN) as proposed by Dai et al. [15]. Incorporating a trainable offset and enlarging the receptive field allows DCN to adeptly accommodate the non-rigid deformation inherent in behaviors.

For an input feature map denoted as *X*, DCN implements a sampling procedure using a 3 × 3 convolutional kernel R,R∈(−1,−1),(−1,0),…,(1,1). Equation (Equation 1) illustrates the calculation process for the output feature map *Y* at position $p_{0}$.
(1)$$Y\left(p_{0}\right)=\sum_{p_{n}\in R}\omega \left(p_{n}\right)•X\left(p_{0}+p_{n}+\Delta p_{n}\right)$$
where $\omega (p_{n})$ represents the weight of the sampling point, $p_{0}$ indicates each position in the input feature map, $p_{n}$ represents all the sample points within *R*, and $\Delta p_{n}$ signifies the offset of these points. The implementation of $\Delta p_{n}$ allows for the adjustment of key element sampling positions, thus enhancing the model’s capability to adapt the behavior recognition of pigs.

DCN generates sampling weights $\omega \left(p_{n}\right)$ at different positions via the offset $\Delta p_{n}$ within each convolution window. However, a limitation lies in obtaining the sampling weights solely through a convolution operation. This single mode of computation restricts DCN’s capability to allocate sampling weights per sampling point dynamically which in turn diminishes the model’s adaptability to the behavior recognition of pigs. To address this, we devised Multi-Path Coordinate Attention (MPCA), inspired by coordinate attention (CA) [26], and the structural details are depicted in Figure 2a. We process input feature $F\in R^{C\times H\times W}$ differently from CA by introducing an additional branch for Global Average Pooling (GAP) [27], hence retaining the global feature information $F_{gap}$ while performing average pooling in the *X* and *Y* directions. We stimulate feature information fusion and exchange in the *X* and *Y* directions through concatenation and convolution operations to yield $F_{hw}$. Then, the features $F_{h}$ and $F_{w}$ are derived by splitting $F_{hw}$. Further application of convolution and sigmoid operations on $F_{hw}$ results in $F_{hw\_weight}$, which is split to obtain feature weights $F_{h\_weight}$ in the *X* direction and $F_{w\_weight}$ in the *Y* direction. Multiplying these weights by $F_{h}$ and $F_{w}$ gives $F_{x}$ and $F_{y}$, revealing more detailed features in the said directions. Simultaneously, the mean operation results in $F_{hw\_weight}$, which when multiplied by $F_{gap}$ yields $\hat{F}$ to mitigate the potential isolation of $F_{gap}$, enabling global information to exchange feature information fully with other branches. The final output feature $F_{output}$ is obtained by multiplying $\hat{F}$ with the input feature *F* and the other two branch features $F_{x}$ and $F_{y}$. We incorporate MPCA following the initial convolution operation in sampling weight computation for $\omega \left(p_{n}\right)$; consequently, MPCA and DCN fuse to form the DCN-MPCA module, as shown in Figure 2b.

DCN-MPCA is proficient at extracting multi-scale features via deformable convolution, and it further augments the feature extraction capacity and adaptability of deformable convolution to a wide range of pig behaviors through the integration of MPCA. This strategy successfully mitigates issues arising from non-rigid deformation during behavior recognition, thereby markedly enhancing the model’s accuracy.

#### 2.2.2. Gather-and-Distribute (GD) Mechanism

Distinct patterns emerge at different scales while recognizing pig behavior. Conventional models, such as the Feature Pyramid Network (FPN), are often reliant on predefined structures to manage features on multiple scales. Some models in the YOLO series incorporate the FPN for enhanced feature extraction. However, these methods may encounter information loss during the fusion of features, particularly when assimilating data generated at various scales by pigs of different sizes and postures. Our proposed solution involves an innovative GD mechanism. This strategy, an extension of the TopFormer’s concept [28], globally integrates features from multiple levels and incorporates this comprehensive information into more complex levels. This GD mechanism makes information exchange at different levels and scales more efficient. The fusion modules for global and local features enable superior capture and communication of contextual information from pig behavior. The GD mechanism mitigates information loss during transmission, thereby enhancing the traditional FPN structure. Figure 3 illustrates the general structure of the GD mechanism. In this depiction, $B_{2},B_{3},B_{4}$, and $B_{5}$ signify the feature maps extracted by the backbone, serving as inputs to the neck. More specifically, $B_{i}\in R^{N\times C_{Bi}\times R_{Bi}}$, where *N* represents the batch size, *C* denotes the number of channels, and the spatial dimensions are expressed by R=H×W. Moreover, the dimensions of $R_{B2},R_{B3},R_{B4}$ and $R_{B5}$ are correspondingly $R,\frac{1}{2}R,\frac{1}{4}R$, and $\frac{1}{8}R$.

The GD process is conducted by three modules: the Feature Alignment Module (FAM), the Information Fusion Module (IFM), and the Information Injection Module (IIM). Task responsibility for the data gathering process is shared between the FAM and the IFM, while data distribution is handled by the IIM.

FAM. FAM’s primary role is aligning input features of different scales to form homogenous scale features. It involves selecting a standard size for feature alignment, then upscaling smaller features via bilinear interpolation and downsizing larger features by average pooling. This uniform scaling ensures consistent spatial dimension across all features. The final step involves concatenation to generate the aligned feature, $F_{align}$.IFM. IFM takes on the responsibility of fusing the features aligned by FAM, creating global information. It does this by inputting the alignment feature, $F_{align}$, from FAM’s output into multi-layer reparameterized convolutional blocks (RepBlocks) or Transformer blocks. From these operations, the global fusion feature $F_{fuse}$ is derived. $F_{fuse}$ is split into different scales features along the channel dimension to produce $F_{inj\_i}$, which is then fused with corresponding scales features.IIM. IIM fuses the split global information from the IFM output with the input information from the corresponding scale in FAM. This compound is injected into the network model to ensure the efficient distribution of global context information through a self-attention approach. As shown in Figure 4, the inputs are the features of the current scale (Fi) and the global features derived from the IFM output ($F_{inj\_Fi}$). Here, *i* is an integer ranging from 3 to 5, representing various levels. Two types of Convs are used with $F_{inj\_Fi}$, yielding $F_{global\_embed\_Fi}$ and $F_{global\_act\_Fi}$, respectively, and $F_{i\_embed}$ is calculated using a Conv of Fi. Should there be any inconsistency in size during the fusion process, it is rectified using average pooling or bilinear interpolation. We then compute $F_{global\_embed\_Fi}$ and Fi, considering attention to generate the fused feature $F_{att\_fuse\_Fi}$. The resultant information undergoes further screening and merging via RepBlock processing, producing the output feature Xi. The corresponding equations used in this process are as follows:
(2)$$F_{global\_act\_Fi}=resize\left(Sigmoid\left(Conv_{act}\left(F_{inj\_Fi}\right)\right)\right),$$
(3)$$F_{global\_embed\_Fi}=resize\left(Conv_{global\_embed\_Fi}\left(F_{inj\_Fi}\right)\right),$$
(4)$$F_{att\_fuse\_Fi}=Conv_{i\_embed}\left(Fi\right)*F_{inj\_act\_Fi}+F_{global\_embed\_Fi},$$
(5)$$Xi=RepBlock\left(F_{att\_fuse\_Fi}\right).$$

The model’s capacity for recognizing variances is improved by the deployment of two key branches: low-stage Gather-and-Distribute (Low-GD) and high-stage Gather-and-Distribute (High-GD). These distinct branches are optimized for the extraction and fusion of feature maps of differing sizes.

Low-GD. The Low-GD component is structured to encompass Low-FAM, Low-IFM, and IIM. Figure 5 provides a detailed representation of this layout. B2, B3, B4, and B5 feature maps from the backbone output are chosen as input for Low-IFM, with B4 determining the target feature size. After feature alignment, the combined feature $F_{align}$ is derived, which is then inputted into Low-IFM and run through a Repblock. This generates the global features $F_{inj\_B3}$ and $F_{inj\_B4}$. B3 and B4 are combined with these in IIM, yielding P3 and P4. Simultaneously, B5 is retained as P5. Ultimately, the output features of Low-GD P3,P4,P5 are produced in this manner.High-GD. Similarly, High-GD is comprised of High-FAM, High-IFM, and IIM as illustrated in Figure 6. The respective inputs of High-FAM are P3,P4,P5 derived from the output of Low-GD. We target P5 for the final size, downsample P3 and P4, and generate $F_{align}$ following feature alignment. Unlike Low-IFM, in High-IFM, $F_{align}$ is fed into the Transformer block and subsequently partitioned to yield $F_{inj\_P4}$ and $F_{inj\_P5}$. These features are injected into P3 and P4, respectively, producing N4 and N5 while retaining P3 as N3, giving us the output features N3,N4,N5.

The implementation of the GD mechanism enables the model to effectively capture multi-scale features and enhances its potential to converge these features. This results in precise recognition of diverse pig behaviors at various stages. Further, the GD mechanism drastically boosts information transmission efficiency, helping avoid information loss. As a result, the model simultaneously upholds low latency and improves precision in behavior recognition.

#### 2.2.3. DM-GD-YOLO

Due to its optimal balance of speed and accuracy, the YOLO series model is extensively utilized for various recognition tasks. This paper employs YOLOv8, the most recent version of the YOLO model, as the fundamental framework for our model. YOLOv8 is composed of a backbone network, a neck network, and detection heads. The backbone network integrates the C2f module, designed to extract features mirroring the Cross Stage Partial Network. The neck network features the Path Aggregation Network (PAN) [29] and the Feature Pyramid Network (FPN) for efficient fusion of features. Post convolution, the output is forwarded to the detection head for the final resolution of the classification and regression problems.

For the task of pig behavior recognition, the traditional convolution operation in C2f employed to extract image features cannot resolve the challenge of non-rigid deformation due to the fixed nature of the convolution kernel. To address this, DCN-MPCA is fused into C2f, while maintaining the basic structure of the C2f module, all traditional convolutions in the Bottleneck are replaced with DCN-MPCA, leading to the creation of the C2f-DM module as depicted in Figure 7. In the backbone, the final three layers of C2f are altered to C2f-DM. C2f-DM retains the principles of C2f, while also adapting well to non-rigid deformation and enhancing the model’s feature extraction capabilities.

On a different note, in YOLOv8, PAN and FPN can only indirectly fuse non-adjacent feature layer information, leading to slow fusion speed and incomplete fusion of behavioral feature information. To address this, the GD mechanism is activated in the neck, allowing for complete fusion and exchange of behavioral feature information. Therefore, we propose a model DM-GD-YOLO for pig behavior recognition, based on the YOLOv8 framework that integrates attention-enhanced deformable convolution and the GD mechanism. The structure of DM-GD-YOLO is illustrated in Figure 7.

## 3. Results and Analysis

### 3.1. Experiment Environment

An NVIDIA GeForce RTX 4090 graphics card (Nvidia, Santa Clara, CA, USA) with a substantial 24 GB video memory was employed for the experiments reported in this study. For Python package management, Anaconda was deployed. Moreover, the Pytorch deep learning framework was adopted. All experiments were conducted on a 64-bit Ubuntu 20.04 system, outfitted with CUDA 12.1, Python 3.8, and Pytorch 2.1.1. The apt selection of hyperparameters is pivotal to the model training procedure. Details of the hyperparameter settings used in the experiments of this study are shown in Table 3.

### 3.2. Evaluation Metrics

This Study employs precision, recall, mean average precision (mAP), parameters, and floating point operations per second (FLOPs) as evaluation metrics to evaluate the model performance. The equations for these metrics are as follows: (6)$$Precision=\frac{TP}{TP+FP}$$
(7)$$Recall=\frac{TP}{TP+FN}$$
(8)$$AP=\int_{0}^{1}P(R)dR$$
(9)$$mAP=\frac{1}{C}\sum_{i=1}^{C}AP_{i}$$
where true positives (TP) denote the number of instances accurately predicted as positive by the model. Similarly, false positives (FP) represent the count of negative instances inaccurately construed as positive, while false negatives (FN) signify instances where positive samples are inaccurately deemed as negative by the model. Precision, calculated as the ratio of TP to the sum of TP and FP, quantifies the model’s accuracy in positive predictions. Meanwhile, recall, computed as the ratio of TP to the sum of TP and FN, evaluates the model’s ability to capture positive samples. The average precision (AP) constitutes the area beneath the precision–recall (P–R) curve, calculated by integrating this curve. The constant ‘C’ signifies the number of classes, marked as seven in this study. For multi-class issues, it is imperative to compute AP individually for each class and subsequently derive their average to determine mAP, given each class owns a distinct P–R curve.

Parameters refer to the quantity of parameters within the model, indicating its size. Smaller models are more easily deployable in various application scenarios. On the other hand, FLOPs quantify the computational complexity of the algorithm, specifically reflecting the number of floating-point operations required for model forward propagation. Models with lower FLOPs are considered less demanding on hardware conditions.

### 3.3. Ablation Experiment

This study employs ablation experiments to substantiate the impact of unifying the C2f-DM module and GD mechanism on YOLOv8’s recognition performance. A higher precision indicates a reduction in model inaccuracies, thereby decreasing the economic strain on farms owing to predictive errors. The elevated recall certifies the comprehensiveness of pig behavior recognition and limits losses arising from recognition failures. Ablation experiment outcomes are presented in Table 4. An analysis of the data illustrates an improvement of 4.3%, 1.3%, and 2.1% in precision, recall, and mAP, respectively, upon integrating only the C2f-DM into the backbone. This signifies that C2f-DM successfully boosts the model’s recognition precision and mAP by enhancing its feature extraction ability, albeit the recall enhancement lacks prominence. Additionally, GD mechanism optimization in the neck improved the model’s precision, recall, and mAP by 0.3%, 5.6%, and 2.1% respectively. The GD mechanism, despite its slight improvement in precision, upgrades recall drastically by augmenting model feature fusion, hence rendering the model capable of recognizing more complete target behaviors. Incorporating both C2f-DM and GD mechanisms substantially enhances the model’s mAP. Upon synergizing these modules, DM-GD-YOLO shows diverse degrees of improvement in all evaluation metrics compared to the YOLOv8 model, with an increment of 1.2% in precision, 5.2% in recall, and 2.6% in mAP. The design of C2f-DM and GD mechanism also leads to a harmonious enhancement of precision and recall. The results prove the optimization and promotion effect of these two modules on the backbone network and the neck of YOLOv8, thereby fortifying the model’s recognition capabilities.

### 3.4. Comparative Analysis of Model Performance

The precision–recall (P–R) curves for YOLOv8 and the proposed DM-GD-YOLO were plotted with precision and recall on the x-axis and y-axis, respectively, as shown in Figure 8. The area under the P–R curve indicates the AP, where a larger area signifies higher AP values. The Figure illustrates varied improvements in AP across different behaviors. Notable behavior classifications with minor AP changes include sniffing and lying. The abnormal behavior of fence-climbing, already achieving a peak AP of 99% in the baseline model, shows marginal enhancement. In contrast, significant AP disparities are observed in climbing and walking behaviors, with a notable increase of 6.2% and 4.9%,, respectively. Particularly noteworthy is the behavior of climbing, which originally yielded an mAP of 88% but rose to 94.2% with DM-GD-YOLO. Furthermore, fighting behavior experienced a 2.5% enhancement. These improvements can be attributed to C2f-DM’s role in enhancing data extraction for behavior recognition, alongside the GD mechanism’s contribution to integrating behavioral information efficiently, thus reducing valuable information loss. This results in superior performance in recognizing complex abnormal behaviors. The proposed DM-GD-YOLO outperforms YOLOv8 by 2.6% in mAP, affirming its effectiveness in improving pig behavior recognition accuracy and notably excelling in recognizing abnormal behaviors.

The two-dimensional confusion matrix visually compares predicted outcomes against actual results to evaluate the effectiveness of behavior recognition models. The confusion matrices depicting the recognition outcomes of the YOLOv8 and DM-GD-YOLO on the test dataset are presented in Figure 9, respectively. In the comparison between YOLOv8 and DM-GD-YOLO, the latter exhibited superior accuracy in recognizing all seven behaviors. Specifically, DM-GD-YOLO correctly recognized sniffing, walking, lying, kneeling, fighting, fence-climbing, and mounting with an additional 7, 24, 47, 12, 1, 1, and 13 instances, respectively. Notably, the enhancements in the recognition of walking and mounting behaviors by DM-GD-YOLO align with the findings from the P–R curve analysis, while the accurate recognition of fence climbing is already high in the baseline model, DM-GD-YOLO maintains a similar performance level with a slight increase.

### 3.5. Model Visualization Analysis

To further substantiate the effectiveness of our model in recognizing various pig behaviors, we selected images depicting pigs in diverse scenarios and conduct. We conducted a comparative analysis between DM-GD-YOLO and the original YOLOv8 as the baseline, leveraging visualizations to facilitate a more intuitive model comparison and visual explanation. The recognition outcomes are depicted in Figure 10, where Figure 10a represents the raw image, Figure 10b illustrates the recognition results of YOLOv8, and Figure 10c showcases the recognition results of DM-GD-YOLO.

In the first row of Figure 10, YOLOv8 failed to detect the lying pig in a relatively dark corner, distant from the monitoring device; however, DM-GD-YOLO accurately detected the pig. The low-contrast colors in the monitored videos created difficulties for YOLOv8 in adapting to this environment, reducing its ability to recognize pig behavior. Consequently, the model’s mAP improved by 8 percentage points in recognizing sniffing and lying behaviors in this setting. Moving to the second row, the similarity between pigs’ sniffing and walking behavior posed a challenge for the model in differentiation. YOLOv8 incorrectly recognized walking as sniffing, whereas DM-GD-YOLO successfully learned to distinguish between the two behaviors, leading to a 19% increase in mAP compared to YOLOv8. In the third and fourth rows, pig interactions involving physical contact caused non-rigid deformations, making it difficult for models to capture these features and recognize abnormal pig behaviors. YOLOv8 incorrectly recognized mounting behavior as kneeling and showed low mAP in recognizing pig fighting behavior, while DM-GD-YOLO’s use of the DCN-MPCA module significantly improved feature extraction capability and precision, accurately recognizing mounting and fighting behaviors. Similarly, in the fifth row, YOLOv8 failed to recognize the pig lying in the corner and the pig with obvious sniffing behavior in the center of the image, exhibiting lower precision in recognizing kneeling behavior, whereas DM-GD-YOLO excelled in these complex environments. Additionally, DM-GD-YOLO demonstrated superior accuracy in recognizing walking and sniffing behavior in the sixth row. In summary, compared with YOLOv8, DM-GD-YOLO exhibits stronger adaptability in complex environments, significantly improving recognition precision and bounding box accuracy, indicating the robustness of the proposed model.

### 3.6. Comparative Analysis with Other Models

To verify the effectiveness and superiority of DM-GD-YOLO in pig behavior recognition, we enlisted a selection of prominent recognition models utilized in livestock behavior recognition for comparative analysis. The evaluation encompassed well-known models such as EfficientDet, Faster R-CNN, YOLOv7, and YOLOv8, all subjected to the same datasets and assessment criteria, with the outcomes meticulously documented in Table 5.

The analysis presented in Table 5 illustrates that DM-GD-YOLO surpasses EfficientDet and YOLOv7 in performance, while its precision slightly lags behind that of EfficientDet, DM-GD-YOLO excels in recall and mAP, achieving rates of 92.2% and 95.3%, respectively. Furthermore, it outperforms in parameters and FLOPs, occupying only 6 MB and 10 G, respectively. Despite a slight increase of 3 MB and 1.9 G in parameters and FLOPs compared to YOLOv8, DM-GD-YOLO demonstrates superior precision, recall, and mAP. This significant enhancement in accuracy is achieved with minimal impact on model size and computational expenses. Unlike the two-stage recognition model Faster R-CNN, DM-GD-YOLO showcases remarkable improvements in precision and mAP, with enhancements of 19.7% and 4.9%, respectively, while maintaining similar recall levels. It also outperforms Faster R-CNN by a significant margin in terms of parameters and FLOPs. Notably, DM-GD-YOLO attains the highest mAP among the considered models at 95.3%.

## 4. Discussion

The welfare breeding of pigs has garnered significant attention in current research. It has been observed that pigs display a range of behaviors, particularly when housed together in high density, leading to an increased likelihood of fighting due to environmental and spatial constraints. Such conflict can result in physical harm, weight loss, and even fatal injuries [30]. Failure to promptly recognize and address abnormal behavior can have serious repercussions, including injuries, stress, and the potential for escape attempts. Thus, abnormal behavior may signal underlying health issues, underscoring the importance of timely and accurate recognition to enable early intervention by farmers, ensuring the well-being of the pigs and minimizing financial losses.

Several studies have utilized wearable devices, such as sensors, to acquire various physiological data of pigs, including acceleration and body temperature. The gathered data was then analyzed extensively to discern and categorize pig behaviors [31]. Chen et al. [32] employed a depth sensor to detect aggressive behavior in pigs, achieving an accuracy of 96.8%. Nonetheless, sensor-based contact behavior recognition methods unavoidably impact the physiological state of pigs, entailing high costs and loss rates. In our research, a non-contact approach leveraging computer vision and deep learning was utilized to recognize pig behaviors. This non-contact recognition technology accurately recognized numerous pig behaviors. Wei et al. [33] proposed a deep learning-based recognition method to capture pig movement and aggressive behaviors such as head-to-head contact, head-to-body contact, neck biting, body biting, and ear biting during fighting. Yang et al. [34] employed Faster R-CNN to recognize feeding behavior in pigs with a precision of 99.6% and a recall of 86.93%. However, the models developed in these studies do not comprehensively cover pig behavior, resulting in farmers not having timely access to all pertinent information. By integrating the DCN-MPCA module and GD mechanism, the DM-GD-YOLO model enhances feature extraction and fusion, enabling the simultaneous recognition of seven pig behaviors, including four common and three abnormal behaviors, thus achieving a comprehensive recognition of pig behaviors and better addressing breeding management needs.

While we have successfully developed the DM-GD-YOLO for pig behavior recognition, it is essential to acknowledge the limitations of this study. Firstly, in real breeding environments with high pig density, the behavior of distant pigs is prone to occlusion by obstacles, introducing uncertainty in feature extraction and behavior recognition. Secondly, although the DCN-MPCA module and GD mechanism in DM-GD-YOLO enhance feature extraction and fusion, they also increase network parameters and complexity, potentially hindering deployment on mobile terminals. Future research will focus on addressing these challenges by enhancing model robustness to the environment and exploring methods to prune the model for greater compactness and efficiency.

Furthermore, it is worth noting that numerous studies on pig abnormal behavior recognition utilize a collection of frames extracted from short videos, combined with time-dimensional features of the behavior, to comprehensively recognize specific individual abnormal behaviors. Currently, our designed video frame recognition model for instantaneous behavior cannot fully leverage contextual information. Our labeling process is based on expert judgment of complete dynamic behaviors captured in the videos, encompassing both intermediate and incomplete behavior forms. Consequently, the model can identify each frame within the behavior process. The model is also aimed at swiftly and effectively capturing transient behavioral features, extracting and fusing them to inform decision-making. It ensures that upon detection of the first frame of abnormal behavior, an immediate reminder message is dispatched to the breeding personnel, facilitating timely intervention to prevent potential hazards from escalating. However, we acknowledge that single-frame recognition may have lower stability compared to video-based multi-frame recognition. This signifies a crucial area for future optimization of the method we have developed.

## 5. Conclusions

This study aims to discern the varied behaviors of pigs in natural breeding environments, with a particular emphasis on health-related abnormal behaviors. To this purpose, we proposed a DM-GD-YOLO recognition model based on the YOLOv8 framework. First, to address the problem of non-rigid deformation commonly associated with pig movements, we integrate the DCN-MPCA module. This technique not only enhances the model’s feature extraction capability by applying an offset during training but also outperforms conventional DCNs in offset weight calculation via the specifically designed MPCA module. We subsequently embed the DCN-MPCA module into C2f, thereby substantially improving the base network’s feature extraction capacity through an optimized C2f-DM. We then ensconced a GD mechanism into the neck, effectively addressing the issue of non-adjacent layer feature fusion of the YOLO network and substantially increasing the network’s information exchange and feature infusion capabilities. We conducted our experiments on the comprehensive self-built dataset of pig behavior, focusing on fattening pigs aged 70–150 days. The experimental findings indicate that DM-GD-YOLO outperforms YOLOv8 regarding precision, recall, and mAP, showing increases of 1.2%, 5.2%, and 2.6%, respectively, while the parameters and FLOPs only increase by 3.0 MB and 1.9 G. Our model was evaluated in pig pens accommodating about 30 pigs. Breeding scenarios surpassing 30 pigs in each pen or employing different trough designs were not investigated, thus necessitating further research. Compared to other prominent recognition models such as YOLOv7, EfficientDet, and Faster R-CNN, the mAP of DM-GD-YOLO improved by 2%, 1.8%, and 4.9%, reaching 95.3%, with the smallest parameters and FLOPs at 6.0 MB and 10.0 G. The adoption of DM-GD-YOLO overcomes the limitations of most existing models by enabling simultaneous recognition of multiple behaviors, improving model feature extraction and fusion capabilities, and facilitating timely detection and management of abnormal pig behaviors, thus reducing potential economic risks. Furthermore, deploying DM-GD-YOLO allows farm staff to intuitively count the frequency of abnormal behavior in the pig herd, providing crucial data support for assessing pig herd health status and enhancing understanding of factors affecting pig welfare. Overall, DM-GD-YOLO not only enhances pig behavior recognition technology but also contributes to the advancement of modern smart farming.

## Figures and Tables

**Figure 1 animals-14-01316-f001:**
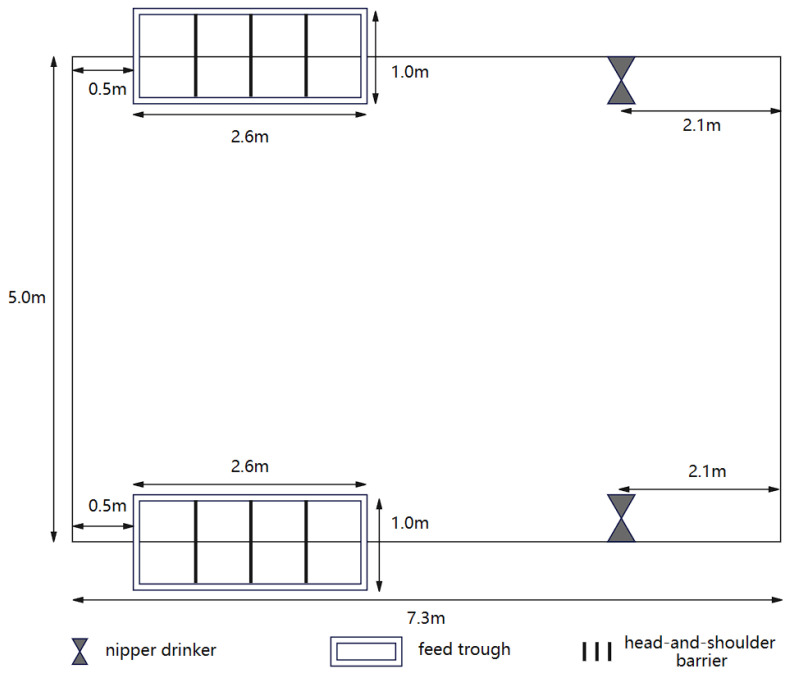
The structure of the pen.

**Figure 2 animals-14-01316-f002:**
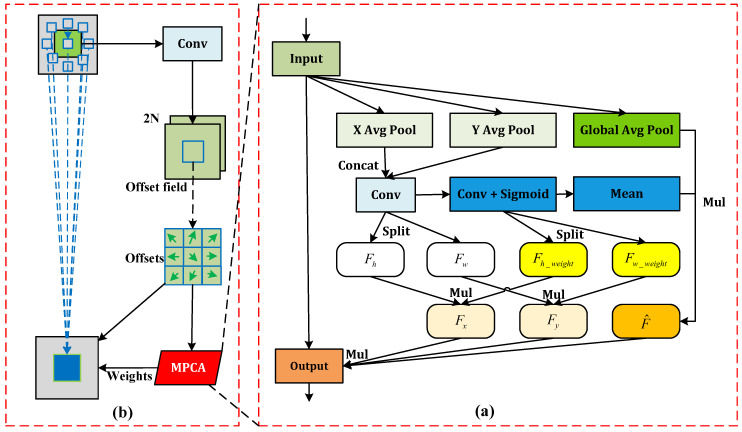
(**a**) The MPCA module; (**b**) The architecture of the DCN-MPCA.

**Figure 3 animals-14-01316-f003:**
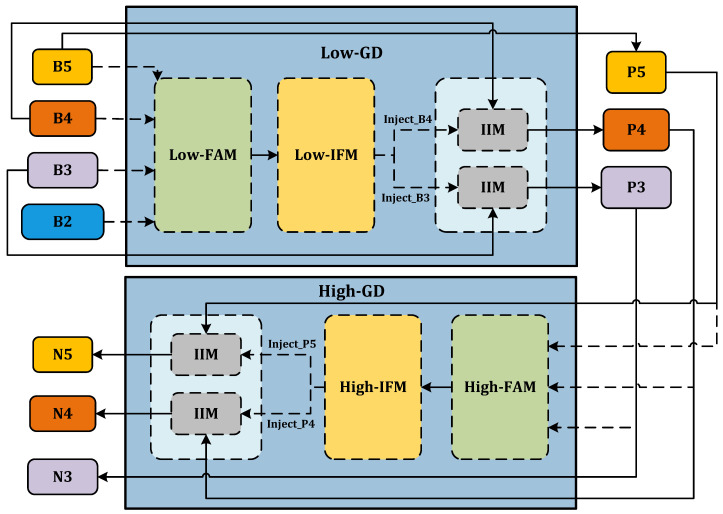
Gather-and-Distribute mechanism.

**Figure 4 animals-14-01316-f004:**
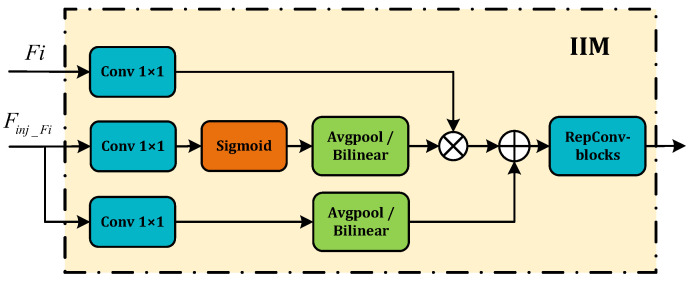
The IIM module.

**Figure 5 animals-14-01316-f005:**
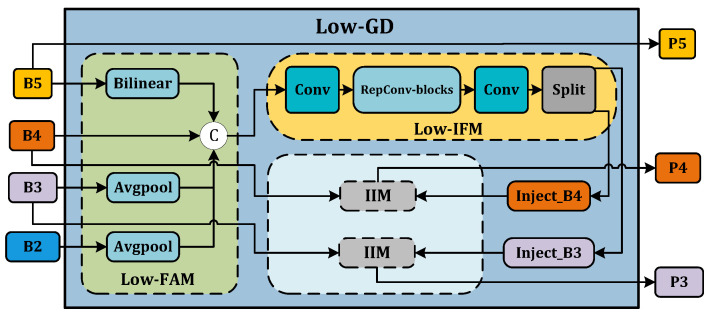
Low-GD branch.

**Figure 6 animals-14-01316-f006:**
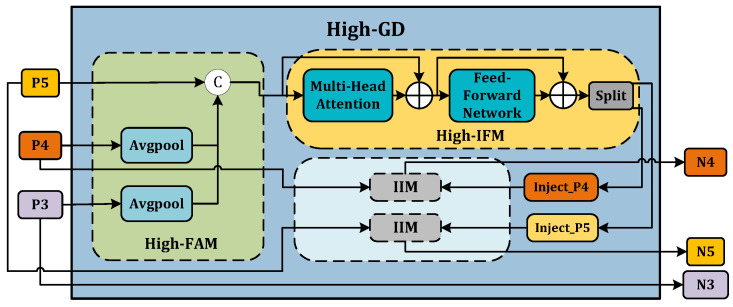
High-GD branch.

**Figure 7 animals-14-01316-f007:**
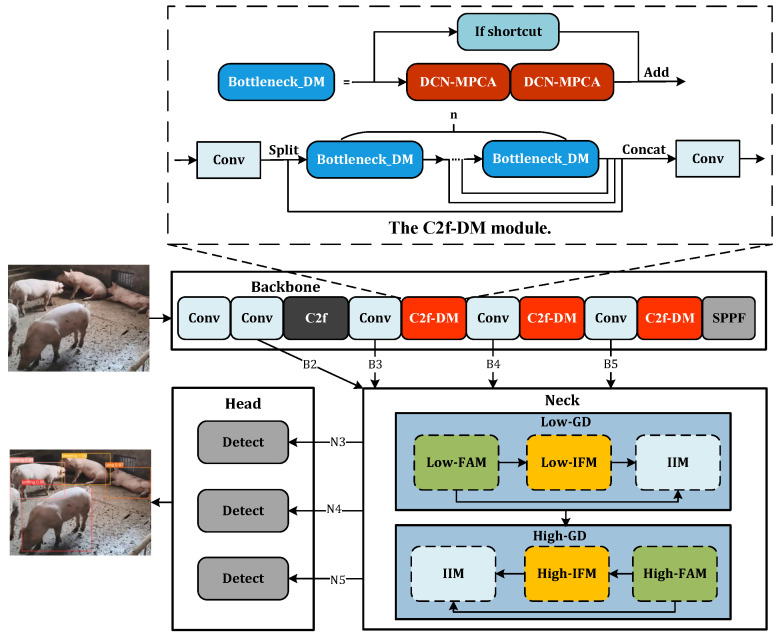
The architecture of the proposed DM-GD-YOLO.

**Figure 8 animals-14-01316-f008:**
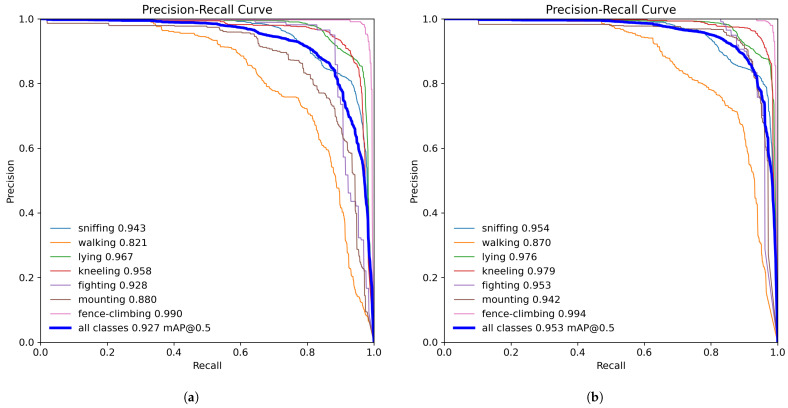
(**a**) P–R curve of YOLOv8; (**b**) P–R curve of DM-GD-YOLO.

**Figure 9 animals-14-01316-f009:**
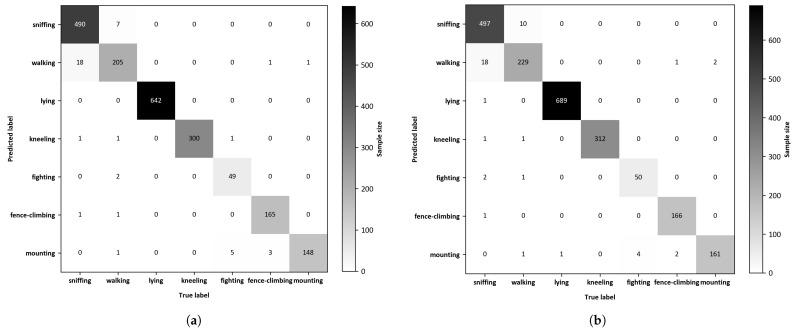
(**a**) Confusion matrix of the YOLOv8; (**b**) Confusion matrix of the DM-GD-YOLO.

**Figure 10 animals-14-01316-f010:**
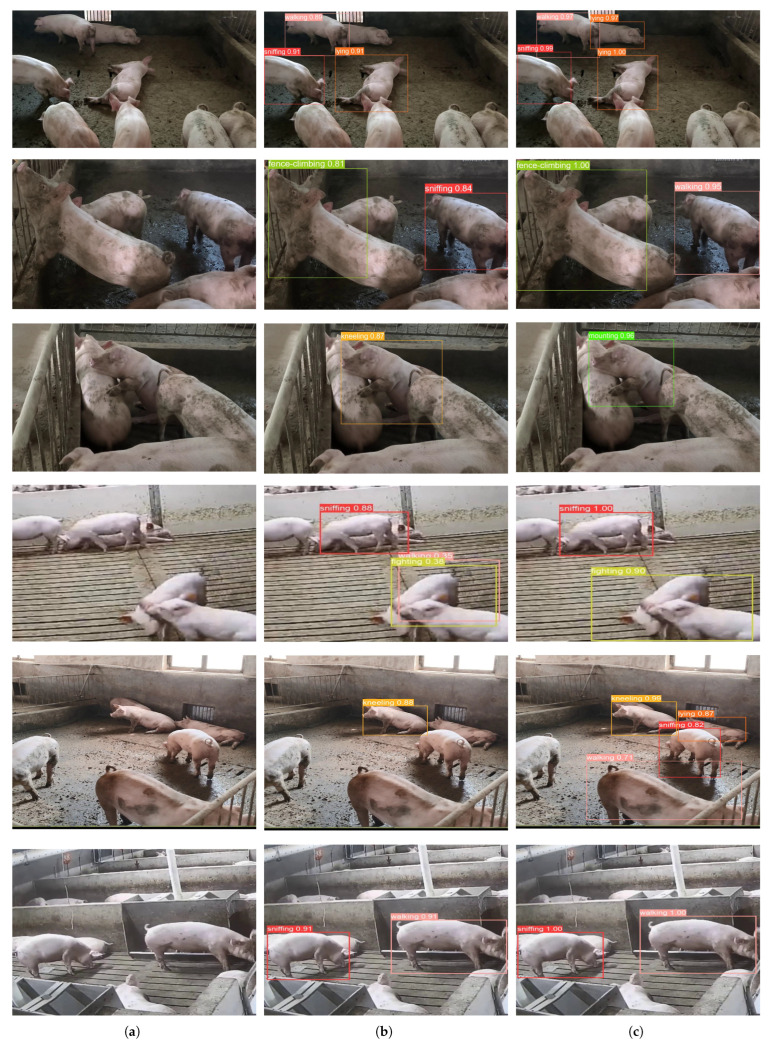
Visualization map: (**a**) Raw Image; (**b**) YOLOv8; (**c**) DM-GD-YOLO.

**Table 1 animals-14-01316-t001:** Recognition rules of typical behaviors of pigs.

Typical Behaviors	Description	Example
Sniffing	Pigs use their snouts to approach or touch objects.	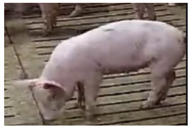
Lying	Pigs lie on the ground with the sternum and udder touching the ground.	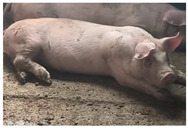
Walking	Pigs move by alternately lifting and landing the front and back legs while using all four legs for support.	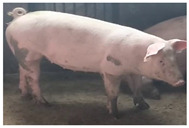
Kneeling	Pigs are supported by their hips and extended front legs, with the hips making contact with the ground.	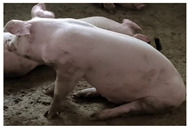
Fighting	Pigs interact by swiftly pushing each other’s neck, head, or ears with their heads.	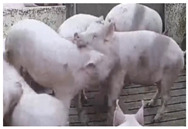
Fence climbing	Pigs place their front legs on the fence, tilting their bodies or positioning them perpendicularly to the ground.	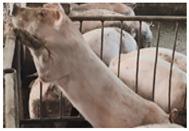
Mounting	Pigs place their two front legs on their partner’s front or back, with or without pelvic insertion.	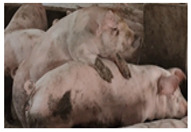

**Table 2 animals-14-01316-t002:** Dataset of pig behavior.

Behavior	Instances of the Training Set	Instances of the Validation Set	Instances of the Test Set	Total
Sniffing	3692	1033	536	5261
Lying	4803	1348	709	6860
Walking	1884	557	264	2705
Kneeling	2126	612	320	3058
Fighting	482	129	55	666
Fence-climbing	1138	351	173	1662
Mounting	1269	351	168	1788

**Table 3 animals-14-01316-t003:** The training parameters for the experiments.

Hyperparameters	Value
Optimization	SGD
Learning rate	0.01
Momentum	0.937
Weight decay	0.0005
Batchsize	64
Warm-up round	3
Warm-up momentum	0.8
Warm-up deviation	0.1
Intersection over Union	0.7
Training epoch	200

**Table 4 animals-14-01316-t004:** Result of ablation experiments based on DM-GD-YOLO.

Model	C2f-DM	GD Mechanism	Precision (%)	Recall (%)	mAP50 (%)
YOLOv8			87.0	87.0	92.7
YOLOv8+C2f-DM	✔		91.3	88.3	94.8
YOLOv8+GD		✔	87.3	92.6	94.8
DM-GD-YOLO	✔	✔	88.2	92.2	95.3

Note: ‘✔’ indicates that a corresponding improvement has been made.

**Table 5 animals-14-01316-t005:** The results of behavior recognition in pigs for different models.

Model	Precision (%)	Recall (%)	mAP (%)	Parameters (MB)	FLOPs (G)
EfficientDet	90.5	88.1	93.5	6.6	11.6
Faster R-CNN	68.5	92.6	90.4	136.8	401.8
YOLOv7	85.9	91.0	93.3	36.6	103.3
YOLOv8	87.0	87.0	92.7	3.0	8.1
DM-GD-YOLO	88.2	92.2	95.3	6.0	10.0

## Data Availability

The dataset will be published at https://github.com/dreamfaker/pig-behavior-dataset (accessed on 17 March 2024) in the future.

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
