# Peer review of "An Integrated Gather-and-Distribute Mechanism and Attention-Enhanced Deformable Convolution Model for Pig Behavior Recognition"

_animals, 2024, doi:10.3390/ani14091316_

Round 1

Reviewer 1 Report

Comments and Suggestions for Authors

In my opinion, instead of writing about what was done in the research study, it would be better to write what the purpose of the research was. In lines 80-81 the authors wrote "...this study proposed a novel method...". Perhaps it would be better to write that the purpose of the research study was to develop a proposal for a new method... I also suggest writing what was the cognitive (scientific) goal and what was the utilitarian (useful) goal of the planned research study. Before stating the purpose of the research study, it would be worth formulating the research problem. I think that based on the review of the state of knowledge presented in the Introduction of the article, a research problem can be easily formulated. I suggest that in the summary of the state of knowledge review, you simply write the sentence: "The research problem is...". Moreover, the research problem can be associated with presenting a gap in the current state of knowledge. This can also be done on the basis of the extensive knowledge presented in the Introduction regarding the assessment of pig behavior, behavior assessment models and others. The last sentence in the Introduction outlines the advantages of a new approach to assessing pig behavior. I think that it would be good to write about the advantages and other important features at the end of the article, when, as a result of the research conducted, it will be possible to indicate what benefits result from implementing a new method of monitoring pig behavior and how it translates into an improvement in the assessment of the animal herd.

I don't understand the phrase "modern pig welfare" (line: 9). Can pig welfare be modern? Maybe it would be worth expressing the idea that the authors wanted to present in a sentence in a different way.

In the Materials and Methods section, I did not find any information about the herd of assessed pigs. How many animals were in the assessed/tested herd? What was the weight of pigs (mean ± SD) in the study group? How old were the pigs (in what age range) included in the study? What breed were the pigs observed in the experiment?

Aren't some of the pig behaviors (in question) specific to a certain age (age range) of animals? Therefore, do the behaviors highlighted in Table 1 apply to all age groups of animals? It would be worth taking up this thread when formulating assumptions for conducting research and in discussing the results of the research study.

Were the animals provided with appropriate conditions during the study to achieve a high level of welfare? Were animal welfare standards met in livestock facilities (piggeries). It is worth developing such information in a separate paragraph in the article. What was the area (in m2) per animal in the pen? Did this unit area meet welfare standards? Were the requirements regarding ventilation (air exchange), internal temperature in the piggery, continuous access to feed and water met? Such information should be included in the description of the livestock facility. 

I would like to know whether the study allowed for the interpretation of intermediate or incomplete forms of pig behavior presented in Table 1? Whether the duration of individual behaviors was measured to determine intensity indicators of the considered animal behaviors. How did these times compare to the total observation time? It would be worth developing these issues in the article.

If the authors included certain indicators in the analysis, e.g. "precision", it would be worth providing a definition of "precision". It would be worth providing this definition regardless of the formula that was quoted in the text of the article (line: 290). The same (request for definition) also applies to "recall" and other indicators.

I don't understand the last sentence in the Conclusions chapter. The authors wrote that "... DM-GD-YOLO ... provides an essential technological perspective for ensuring pig welfare...". I am not sure whether DM-GD-YOLO can ensure the welfare of pigs. I think that DM-GD-YOLO can be a tool to help improve knowledge and recognize factors identifying pig welfare.

Author Response

Comments 1: In my opinion, instead of writing about what was done in the research study, it would be better to write what the purpose of the research was. In lines 80-81 the authors wrote "...this study proposed a novel method...". Perhaps it would be better to write that the purpose of the research study was to develop a proposal for a new method...

Response 1: Thank you for bringing this to our attention. We have revised our description in the introduction section by incorporating a comprehensive description of the purpose of the research. (Line 83~84)

Comments 2: I also suggest writing what was the cognitive (scientific) goal and what was the utilitarian (useful) goal of the planned research study.

Response 2: Thanks for the comments. The study now encompasses the cognitive and utilitarian objectives in the introduction. The cognitive objective aims to achieve precise recognition of both common and abnormal pig behavior. The utilitarian objective seeks to assist farmers in promptly identifying potential risks to pig health, consequently minimizing economic losses. (Line 85~86)

Comments 3: Before stating the purpose of the research study, it would be worth formulating the research problem. I think that based on the review of the state of knowledge presented in the Introduction of the article, a research problem can be easily formulated. I suggest that in the summary of the state of knowledge review, you simply write the sentence: "The research problem is...". Moreover, the research problem can be associated with presenting a gap in the current state of knowledge. This can also be done on the basis of the extensive knowledge presented in the Introduction regarding the assessment of pig behavior, behavior assessment models and others.

Response 3: Thanks for the comments. The introduction section now includes the research problem before stating the purpose of the research study, along with an exposition of gaps in the current state of knowledge. The research problem pertains to the deficiencies in the existing model, which struggles with simultaneous recognition of various pig behaviors, particularly abnormal ones, and lacks robust feature extraction and fusion capabilities, leading to potential information loss (Line 80~83).

Comments 4: The last sentence in the Introduction outlines the advantages of a new approach to assessing pig behavior. I think that it would be good to write about the advantages and other important features at the end of the article, when, as a result of the research conducted, it will be possible to indicate what benefits result from implementing a new method of monitoring pig behavior and how it translates into an improvement in the assessment of the animal herd.

Response 4: Thank you for your valuable suggestion. We have described the advantages and benefits of our approach and elaborated in the conclusion on how it contributes to the enhancement of livestock health. (Line 501~509)

Comments 5: I don't understand the phrase "modern pig welfare" (line: 9). Can pig welfare be modern? Maybe it would be worth expressing the idea that the authors wanted to present in a sentence in a different way.

Response 5: Thank you for pointing this out. We are very sorry for the mistake in the previous version of the manuscript and have revised our description. (Line 8~10)

Comments 6: In the Materials and Methods section, I did not find any information about the herd of assessed pigs. How many animals were in the assessed/tested herd? What was the weight of pigs (mean ± SD) in the study group? How old were the pigs (in what age range) included in the study? What breed were the pigs observed in the experiment?

Response 6: Thank you for your valuable suggestion. Related information about the herd of assessed pigs has been added to the data sources section, including number, weight, age and  breed. (Line 112~114)

Comments 7: Aren't some of the pig behaviors (in question) specific to a certain age (age range) of animals? Therefore, do the behaviors highlighted in Table 1 apply to all age groups of animals? It would be worth taking up this thread when formulating assumptions for conducting research and in discussing the results of the research study.

Response 7: Thanks for the comments. Upon video collection, we observed the seven behaviors in Table 1 across various age ranges of all the tested fattening pigs, indicating that these behaviors are not restricted to a specific age range. Simultaneously, enhancements have been made to the data description. (Line 23, Line 114 and Line 402~403)

Comments 8: Were the animals provided with appropriate conditions during the study to achieve a high level of welfare? Were animal welfare standards met in livestock facilities (piggeries). It is worth developing such information in a separate paragraph in the article. What was the area (in m2) per animal in the pen? Did this unit area meet welfare standards? Were the requirements regarding ventilation (air exchange), internal temperature in the piggery, continuous access to feed and water met? Such information should be included in the description of the livestock facility.

Response 8: Thank you for your valuable suggestion. The conditions of the livestock facilities, all of which meet the pig welfare standards, have been revised in the data source section. Additionally, we have depicted a figure illustrating the pig pen structure. (Line 115~124, Figure 1.)

Comments 9: I would like to know whether the study allowed for the interpretation of intermediate or incomplete forms of pig behavior presented in Table 1? Whether the duration of individual behaviors was measured to determine intensity indicators of the considered animal behaviors. How did these times compare to the total observation time? It would be worth developing these issues in the article.

Response 9: Thanks for the comments. The behavior descriptions in Table 1 based on reference 23 and suggestions from breeder. The behaviors of sniffing, lying, kneeling, and fence-climbing exhibit minimal variation over time, while walking, fighting, and mounting behaviors are characterized by their dynamic nature in Table 1. Our labeling process is based on expert judgment of complete dynamic behaviors captured in the videos, encompassing both intermediate and incomplete behavior forms. Consequently, the model can identify each frame within the behavior process. The primary aim of this study is to accurately detect abnormal behavior occurrences and promptly alert breeders, enabling timely intervention to prevent potential risks from escalating. Notably, we have not yet conducted time-based statistical analysis of behavior occurrence durations, a noteworthy aspect that requires tracking technology support and presents a focal point for future enhancements in our project. Therefore, we have included references to the limitations of our approach in the discussion section. (Line 465~478)

Comments 10: If the authors included certain indicators in the analysis, e.g. "precision", it would be worth providing a definition of "precision". It would be worth providing this definition regardless of the formula that was quoted in the text of the article (line: 290). The same (request for definition) also applies to "recall" and other indicators.

Response 10: Thanks for the comments. The definitions of all evaluation indicators have been added to the evaluation metrics section. (Line 316~319 and Line 324~327)

Comments 11: I don't understand the last sentence in the Conclusions chapter. The authors wrote that "... DM-GD-YOLO ... provides an essential technological perspective for ensuring pig welfare...". I am not sure whether DM-GD-YOLO can ensure the welfare of pigs. I think that DM-GD-YOLO can be a tool to help improve knowledge and recognize factors identifying pig welfare.

Response 11: Thanks for the comments. We have amended the description of the role of DM-GD-YOLO in the conclusion section. (Line 507~509)

Reviewer 2 Report

Comments and Suggestions for Authors

This manuscript developed a tailored YOLOv8 model, incorporating deformable convolution with multi-head attention, and gather-and-distribution mechanisms to detect seven typical behaviors of pig: 4 common (sniffing, walking, lying, and kneeling) and 3 abnormal (fence-climbing, mounting, and fighting). The model showed better performance than standard object detection models. The manuscript is well structured and written with enough explanation. One of the major contributions of this study is that the authors are going to publish their datasets for other researchers, which is highly commendable and recommend publishing them once the paper is accepted. However, I have a few remarks on it, please consider them before publication.

1.     Regarding abnormal behavior, considering only a frame how can we confirm the exact behavior. If you read some abnormal behavior related literature, they used a short episode of frames to confirm them. Please verify it and mention in the discussion section how we can detect such behavior from single frame detection.

2.     Thoroughly check the manuscript and correct some typos like in Table 5 Rfficient,…

3. Although the author claimed they achieved very good results from their proposed model DM-GD-YOLO, Fig. 9 showed that the model undetected many partially occluded pigs. Why did the model not detect behavior of all pigs except major occluded ones. Please mention the data labeling standards you followed for partially occluded objects in the data preprocessing section to justify it.

Comments on the Quality of English Language

Minor typo checking is required.

Author Response

Comments 1: Regarding abnormal behavior, considering only a frame how can we confirm the exact behavior. If you read some abnormal behavior related literature, they used a short episode of frames to confirm them. Please verify it and mention in the discussion section how we can detect such behavior from single frame detection.

Response 1: Thank you for your valuable suggestion. It is worth noting that numerous studies on pig abnormal behavior recognition utilize a collection of frames extracted from short videos, combined with time dimensional features of the behavior, to comprehensively recognize specific abnormal behaviors. When applying video-based multi-frame recognition methods, the training videos typically feature one or two pigs relevant to the behavior under study. However, when implementing these methods in environments with multiple pigs, it becomes imperative to incorporate target tracking technology to ensure continuity in identifying the pigs involved in the behavior. Currently, our designed video frame recognition model for instantaneous behavior cannot fully leverage contextual information. Our labeling process relies on expert judgment of complete dynamic behaviors captured in the videos, encompassing both intermediate and incomplete behavior forms. Consequently, the model can identify each frame within the behavior process. And the model is aimed at swiftly and effectively capturing transient behavioral features, extracting, and integrating them to facilitate decision-making. Upon detecting the first frame of abnormal behavior, the model promptly sends a notification to the caretakers, allowing timely intervention to prevent potential risks. Despite the benefits, single-frame recognition may exhibit lower stability compared to video-based multi-frame recognition, highlighting an area for future improvement in our project. Meanwhile, we have discussed the limitations of our approach in the discussion section. (Line 465~478)

Comments 2: Thoroughly check the manuscript and correct some typos like in Table 5 Rfficient,…

Response 2: Thank you for pointing this out. We are very sorry for the mistakes in the previous version of the manuscript. In this version, we carefully checked all spelling issues and corrected the errors that occurred.

Comments 3: Although the author claimed they achieved very good results from their proposed model DM-GD-YOLO, Fig. 9 showed that the model undetected many partially occluded pigs. Why did the model not detect behavior of all pigs except major occluded ones. Please mention the data labeling standards you followed for partially occluded objects in the data preprocessing section to justify it.

Response 3: Thank you for your valuable suggestion. During the labeling process, pigs with occluded areas exceeding 30% or with occluded heads were not labeled. Therefore, for instance, in the image located at column c, line 1 of Fig. 10, the heads of all four pigs are occluded at the bottom, making it challenging to discern whether they are sniffing or walking. Similarly, the pig obscured by another pig climbing the fence in column c, line 2, presents a similar challenge. Consequently, instances may occur where occluded pigs cannot be detected. Thanks for the reminder and the criteria for labeling occluded behaviors have been added in the data preprocessing section.(Line 139~142)

Response to Comments on the Quality of English Language

Point 1: Minor editing of English language required.

Response 1:  Thank you for pointing this out. We apologize for the errors in the previous version of the manuscript. In this revised version, we meticulously reviewed and rectified all spelling issues.

Reviewer 3 Report

Comments and Suggestions for Authors

The authors presented a very clear explanation of the pig behaviors with both photos and a written description. This reviewer is concerned that using videos from only one farm may influence identification of different behaviors. A photo of the pens is recommended. Many pig farms have pens with more than 30 pigs in each pen. The question this reviewer is asking is: Would larger pens with more pigs have an effect on problems with occluded images that may lower the programs ability to detect different behaviors? Feed trough design may also influence behavior Pigs will behave differently in a system where all the pigs line up at one time to eat from a trough versus eating dry feed from a feeder. In this system, pigs take turns eating all the feed they want. This reviewer recommends adding to both the abstract and the introduction information about the farm.

The conclusions should be limited to the type of farm you collected the images from. Also state what the feeder design is.

Line 8 - Add a sentence. This study was conducted on a farm that had 30 pigs in each pen.

Line 25 - Add a sentence that this model outperformed other popular models when tested on a farm that had 30 pigs in each pen.

Line 460 - Add a sentence to the conclusions that - Our model was tested on a farm that had 30 pigs in each pen. Additional research may be needed for use on farms that have larger numbers of pigs in each pen or a different feed trough design.

Author Response

Comments 1: The authors presented a very clear explanation of the pig behaviors with both photos and a written description. This reviewer is concerned that using videos from only one farm may influence identification of different behaviors. A photo of the pens is recommended. Many pig farms have pens with more than 30 pigs in each pen. The question this reviewer is asking is: Would larger pens with more pigs have an effect on problems with occluded images that may lower the programs ability to detect different behaviors? Feed trough design may also influence behavior. Pigs will behave differently in a system where all the pigs line up at one time to eat from a trough versus eating dry feed from a feeder. In this system, pigs take turns eating all the feed they want. This reviewer recommends adding to both the abstract and the introduction information about the farm. The conclusions should be limited to the type of farm you collected the images from. Also state what the feeder design is.

Line 8 - Add a sentence. This study was conducted on a farm that had 30 pigs in each pen.

Line 25 - Add a sentence that this model outperformed other popular models when tested on a farm that had 30 pigs in each pen.

Line 460 - Add a sentence to the conclusions that - Our model was tested on a farm that had 30 pigs in each pen. Additional research may be needed for use on farms that have larger numbers of pigs in each pen or a different feed trough design.

Response 1: Thanks for the comments. It is essential to note that our model has not yet undergone testing in pen with more than 30 pigs. With an increase in the number of pigs within a pen, occlusion occurrences are more frequent, potentially impacting pig behavior. Additionally, diverse feed trough designs may also influence pig behavior. Thus, we have defined the method's scope of application in this paper. The design of feed trough has been described in the data sources section, and we have depicted a figure outlining the pen’s detailed structure (Line 121~124, Figure. 1). Lastly, we have meticulously incorporated and revised the corresponding description (Line 6, Line 27 and Line 406~407) based on your suggestions.

Round 2

Reviewer 3 Report

Comments and Suggestions for Authors

Manuscript ID 2946326 - An integrated gather and distribute mechanism and attention enhanced deformable convolution model for pig behavior recognition should be accepted for publication.

The authors have made the revisions I recommended and the revised paper is acceptable for publication in its present form.